# Improving effective contraception uptake through provision of bridging contraception within community pharmacies: findings from the Bridge-it Study process evaluation

Susan Patterson [1], Lisa McDaid [1,2] Kristina Saunders,[3] Claire Battison,[4] Anna Glasier,[5] Andrew Radley [6,7] Judith M Stephenson,[8] Anne Johnstone,[5] Alessandra Morelli [9] Deirdre Sally,[10] Nicola Stewart,[10] Sharon Tracey Cameron [5,11]

For numbered affiliations see end of article.

**Correspondence to**
Susan Patterson;
susan.patterson@glasgow.ac.uk

## ABSTRACT

**Objective** To present process evaluation results from the Bridge-it Study, a pragmatic cluster randomised cross-over trial to improve effective contraception uptake through provision of the progestogen only pill (POP) plus sexual and reproductive health (SRH) clinic rapid-access to women presenting to community pharmacies for emergency contraception (EC).

**Research design and methods** A multimethod process evaluation was conducted to assess intervention implementation, mechanisms of change and contextual factors. Data were gathered from screening logs (n=599), observations of pharmacist training, analysis of data from 4-month follow-up questionnaires (n=406), monitoring of contemporaneous events and qualitative interviews with 22 pharmacists, 5 SRH clinical staff and 36 study participants in three participating UK sites in Lothian, Tayside and London.

**Results** The intervention was largely delivered as intended and was acceptable. Pharmacists', SRH clinical staff and participants' accounts highlighted that providing a supply of POP with EC from the pharmacy as routine practice may have positive impacts on contraceptive practices in the short term, and potentially longer term. Key mechanisms of change included ease of access, increased awareness of contraception and services, and greater motivation and perceptions of self-efficacy. Few participants took up the offer to attend an SRH service (rapid-access component), and existing barriers within the SRH context were apparent (eg, lack of staff). Participant accounts highlight persistent barriers to accessing and using routine effective contraception remain.

**Conclusions** Implementation appeared to be acceptable and feasible, highlighting the potential for provision of POP within EC consultations as routine practice in community pharmacies. However, lack of engagement with the rapid access component of the intervention and existing barriers within the SRH context suggest that signposting to SRH services may be sufficient. Wider implementation should consider ways to address key implementation challenges to increase effectiveness and sustainability, and

### Strengths and limitations of this study

► The Bridge-it study process evaluation combined qualitative and quantitative methodologies to provide comprehensive and robust insights into implementation of the intervention, mechanisms of change and important contextual factors.

► Due to participants being followed up 4-month postintervention, and qualitative interviews taking place at one time point, we are unable to comment on continuation of the chosen contraceptive method and longer-term implementation of the service.

► While purposive sampling was employed to ensure providers and participants recruited for interviews were diverse, the generalisability of findings are limited to accounts from those who agreed to take part in the trial, and to those who agreed to be interviewed.

► Providers and study participants were asked to reflect on experiences up to 6 months previously, which may have impacted on recall.

► Due to limited use of the rapid access component and difficulties recruiting sexual and reproductive health (SRH) clinical staff for interview, accounts of experiences within the SRH context were limited; additionally, due to practical reasons, we were unable to observe implementation of the intervention within the pharmacy or SRH context, making assessing fidelity of the intervention difficult.

to overcome persistent barriers to accessing and using effective contraception.

**Trial registration number** ISRCTN70616901.

## INTRODUCTION

Unintended pregnancy remains a public health issue within the UK, with abortion rates in 2020 reaching the highest numbers recorded since records began (13.4 per

1000 women (aged 15–44) in Scotland[1]; 18.2 per 1000 women (aged 15–44) in England and Wales[2]). Additional outcomes of unintended pregnancy include miscarriage, ectopic pregnancy, unwanted or mistimed birth, all with the potential to have adverse impacts on maternal and child health.[3] Oral emergency contraception (EC) can be used to prevent unintended pregnancy, and is typically accessed through community pharmacies.[4][5] Guidance from the Faculty of Sexual and Reproductive Health (SRH) emphasises the importance of rapid access to ongoing contraception after EC,[6] but many face barriers to accessing further contraception such as difficulties accessing general practitioner (GP) appointments and contraceptive services, fuelled by sexual health service funding cuts, and more recently exacerbated by the coronavirus pandemic.[7] Within this context, pharmacies present a promising venue for increasing access to contraception, with long opening hours and wide geographical coverage,[8][9] but until recently, were only able to provide condoms without a prescription. In July 2021, progestogen-only contraceptive pills were approved for sale over the counter in community pharmacies in the UK[10]), and while this represents a step forward in provision, the requirement to pay may further increase already-evident inequalities in access and outcomes.[1][2] Taking this into consideration, in November 2021, following the successful Bridge-it study trial,[11] women in Scotland are now able to obtain a 3-month supply of the progestogen-only pill free of charge from within community pharmacies.[12]

The Bridge-it Study was a pragmatic cluster randomised cross-over trial designed to determine the effectiveness of a bridging contraceptive service within community pharmacies in increasing uptake of effective contraception. The intervention consisted of the provision of a 3-month supply of the progestogen only pill (POP) (75 µg desogestrel/day) after EC (levonorgestrel 1.5 or 3 mg) at no cost within EC consultations, alongside a study card which on presentation at participating SRH services enabled rapid access to appointments for advice and provision of ongoing contraception. The card provided information on the location and opening times of the participating SRH clinics (three in London, two in Tayside and one in Lothian). In the control arm in which women were not provided with the POP, participants were advised to attend their GP/SRH service or usual contraceptive provider for contraception after EC (standard care). Participants were followed up at 4 months, either by telephone interview with a research nurse, or by self-administered questionnaire via email, and asked about contraceptive use, their experience within the pharmacy, and use of the rapid access card (intervention group). In total, 29 UK pharmacies in London (n=14), Lothian (n=12) and Tayside (n=3) participated in the study, and recruited 636 participants (intervention n=316; control n=320). Analysis of the main outcome of the study demonstrated the effectiveness of the intervention, with a greater proportion of women using effective contraception at 4-month follow-up within

the intervention group (58.4% SD 21.6) compared with the control group (40.5% SD 23.8).[11] Full details on the trial protocol and outcomes are reported elsewhere.[11][13]

This paper reports a multi-method process evaluation of the Bridge-it intervention, included to assess implementation, mechanisms of change and context (eg, external factors that may influence implementation and effectiveness), in order to better understand the overall intervention outcomes and shed light on reasons why the intervention was effective (or not).[14] The process evaluation was underpinned by a conceptual framework, which incorporated a range of causal assumptions, and acknowledgement of the potential impact of contextual factors on achievement of key outcomes (see figure 1). Formative research highlighting desire among women presenting for EC at community pharmacies for access to ongoing contraception through community pharmacies, and existing barriers to access faced in more traditional settings,[15][16] informed the design of the process evaluation, which aimed to understand:

► Was the intervention implemented as planned?
► How did the delivered intervention impact on contraceptive practices?
► How did the local and broader context affect implementation and outcomes?

Given the recent changes in POP availability within pharmacies in Scotland, this paper is timely, and will help to shed light on key issues and how wider implementation of the service within community pharmacies may be optimised.

## METHODS

The process evaluation used an evaluation framework to allow the systematic synthesis of data on implementation, perceived mechanisms of change, and the impact of context on implementation and outcomes (see figure 1). The funder had no role in the intervention or evaluation design.

### Data sources and analysis
#### Qualitative interviews with pharmacists, SRH clinical staff and participants

Qualitative data were collected from those delivering the intervention (pharmacists and SRH clinical staff), and those receiving it (Bridge-it Study participants). Semi-structured qualitative interviews were conducted by telephone by the process evaluation research assistants (SP and KS), who were not involved in the development or implementation of the main trial, and had no relationship with providers or study participants. Topic guides were specific to each group (see online supplemental data file 1), exploring issues such as acceptability of the intervention, experiences of delivering the intervention or of receiving it, impacts on contraceptive practices, and contextual issues relevant to implementation and outcomes. Consent was obtained, interviews were audiorecorded, transcribed verbatim, anonymised and

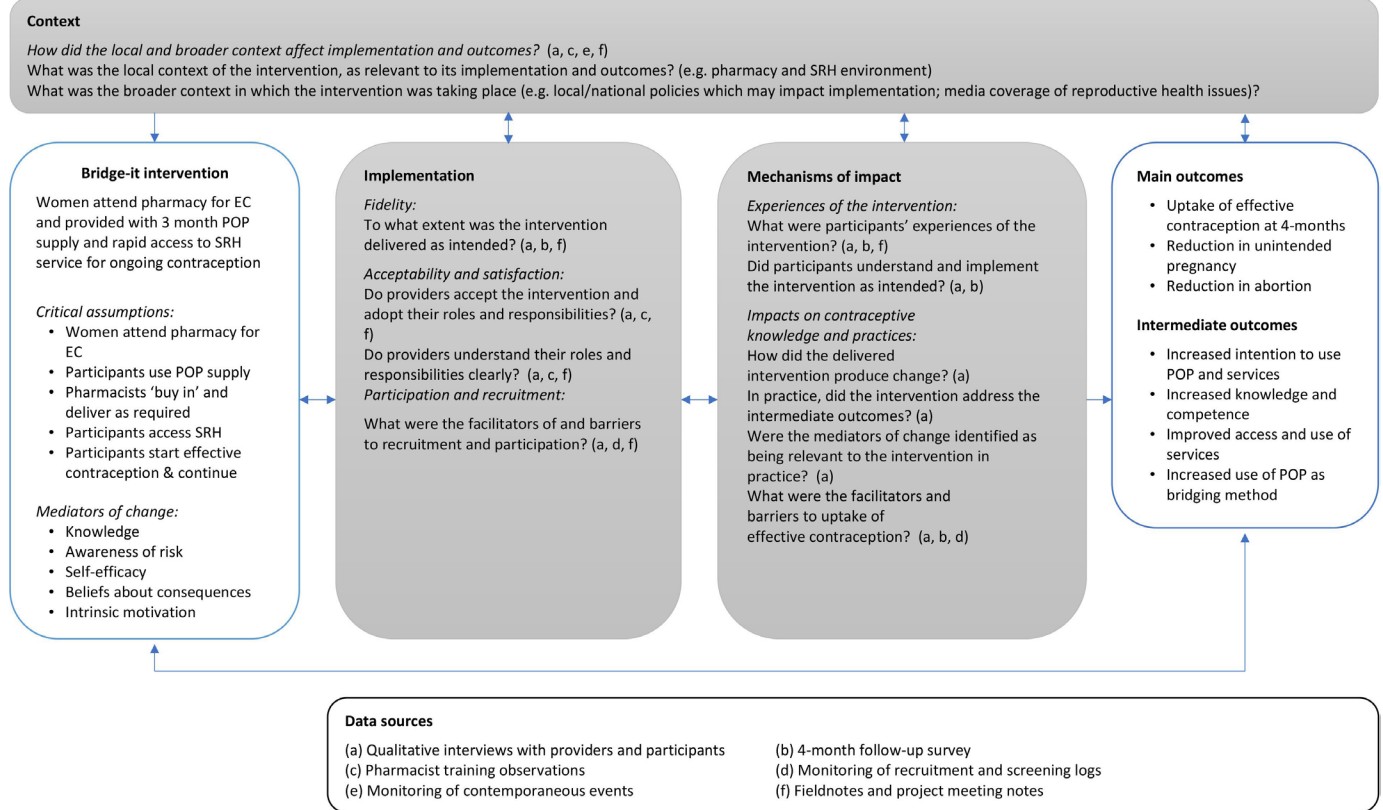

**Figure 1** The Bridge-it study process evaluation framework. EC, emergency contraception; POP, progestogen only pill; SRH, sexual and reproductive health.

uploaded to QSR NVivo V.10 for analysis. Data analysis was undertaken using Framework Analysis, where data are coded, indexed and charted systematically to facilitate synthesis of key themes.[17] The thematic coding framework was developed by the process evaluation team (SP, KS and LM), largely using a deductive approach guided by the research questions, process evaluation framework and topic guide, but also shaped by new themes generated through the familiarisation stage and open coding. This thematic coding framework was used to systematically code and chart the data using constant comparison to ensure all perspectives were represented, and enabled further analysis to shed light on commonalities and differences by themes within and across the data. The framework analysis method was particularly useful for this multi-method process evaluation, as non-interview data could easily be incorporated within matrices (eg, fieldnotes; observational data).

Research nurses asked participants for consent to be contacted for a qualitative interview at the end of the 4-month follow-up questionnaire, and interviews were conducted between November 2018 and October 2019. Purposive sampling was used aiming to recruit a representative and diverse sample, with participants sampled by area, age, ethnicity, use of the study POP, and attendance at SRH. However due to difficulties in recruiting, we approached all participants who agreed to be contacted for interview. In total, 36 intervention participants were interviewed (figure 2), and participant characteristics were largely representative of the main study sample,[11] with similar characteristics to EC users nationally.[15 16] Intervention participants were aged 18–37, and the majority were under 24 (n=21) and described themselves as white (n=29). Many had used EC previously (n=17), over half used all three packets of POP (n=21) and five had attended the SRH clinic. Almost half (n=16) were using a POP or another effective contraceptive method at the time of interview. Most of the interviews were conducted with participants in Edinburgh, reflecting the greater number of participants recruited to the study within Lothian (recruitment began earlier and included more larger chain pharmacies with high EC dispensing rates), as well as lower response to the 4-month follow-up questionnaire, and willingness to be contacted to take part in a qualitative interview among study participants in London. On average interviews lasted between 30 and 60 min.

During training sessions, pharmacists were presented with information about the process evaluation interviews, and later contacted by the Trial manager or research nurse to ask if they were willing to be contacted for an interview by the process evaluation research assistant. The interviews were conducted between July 2018 and July 2019, with most taking place once recruitment had ended within their particular pharmacy. In total, 22 pharmacists were interviewed, 12 from Lothian, three from Tayside and seven from London. The aim had been to interview one pharmacist from each participating pharmacy. The main pharmacies not represented (n=7) are

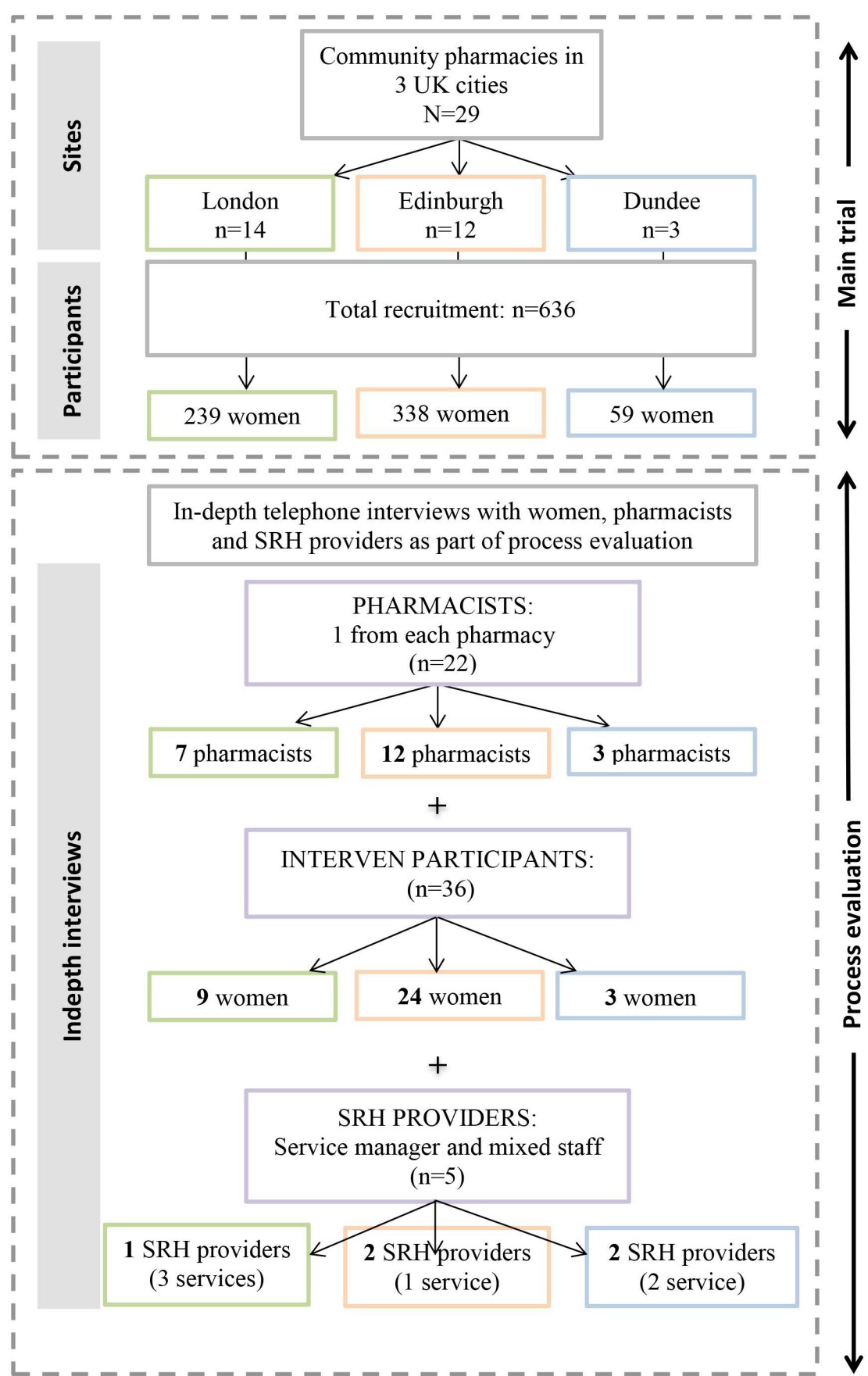

**Figure 2** Breakdown of main study and process evaluation (PE) recruitment and sites. SRH, sexual and reproductive health

based in South London and it had not been possible to conduct interviews before the study was discontinued. Interviews typically lasted 30–45 min.

SRH clinical staff were contacted by the research nurses and asked if they were willing to be contacted for interview, with subsequent interviews conducted between May and October 2019, approximately 4–6 months after study recruitment had ended to allow time for experience of participants attending their service. Five SRH clinical staff were interviewed within three of the participating NHS sites (two in Lothian, two in Tayside and one in London). We had originally aimed to interview 3–4 staff members from each service, however, recruitment was challenging, particularly due to low Bridge-it participant attendance at SRH clinics. Interviews typically lasted 30–45 min.

### Researcher field notes and meeting minutes
Fieldwork reflections were recorded and meeting minutes analysed to explore factors that may have influenced consistency or quality of data and implementation.

### Monitoring of pharmacy recruitment and observations of training
Pharmacy recruitment was monitored using a standardised form to record factors relating to pharmacy selection, including reasons for inclusion/exclusion (eg, location; high EC distribution); and reasons for acceptance/refusal (eg, lack of interest; high workloads) (see online supplemental data file 2). Thirteen Bridge-it training sessions for pharmacists were observed by a research assistant in Scotland, and all intervention and training materials were reviewed. A training observation proforma (see online supplemental data file 3) was completed by the research assistant, with particular attention paid to the way key intervention mechanisms were presented to, and apparently understood by, pharmacists. Written observational data were transcribed into Microsoft word, thematic analysis conducted guided by the proforma, and descriptive summaries written.

### Quantitative data
The process evaluation drew on the baseline questionnaire (demographic details; reproductive history; previous contraceptive use), the 4-month follow-up questionnaire (contraceptive use; experience in pharmacy; use of rapid access card; n=406, 64% of participants) (see online supplemental data file 4), and pharmacist screening logs (n=599), detailing reasons for exclusion/declining. Data were analysed descriptively (software package SPSS V.25).

### Synthesis of multiple data sources
All process evaluation data were analysed prior to reporting of trial outcome data to minimise bias in interpretation, and the process evaluation team regularly discussed analysis progress for each source of data collection, allowing any issues encountered to be resolved. Following independent analysis of each data source, the data were synthesised to address the three key research questions relating to implementation, mechanisms of impact and the role of context. An analytical integration matrix was

created to compare findings from each stage (see online supplemental data file 5). Analysis addressed complementary findings from each source of data and drew out synergistic interpretations to facilitate a broader holistic picture of how the intervention worked in practice.

### Patient and public involvement
Members of the participating Edinburgh SRH service patient and public involvement group were service users and contributed to the design of the Bridge-it study process evaluation through reviewing and commenting on study documentation. Members participated in the trial steering committee to assist with oversight of the study.

## RESULTS
This section presents key findings relating to implementation, mechanisms of impact, and the influence of contextual factors on implementation. Additional findings for each measure are presented in online supplemental data files 6, 7 and 8.

### Implementation: acceptability and fidelity
The intervention was acceptable to pharmacists who saw it as an important way to improve access to contraception and help reduce repeat EC use and unwanted pregnancy rates: 'it shows that people are taking the issue of unwanted pregnancy seriously and they're trying to improve, you know, the accessibility of services to women' (Pharmacist 18, Lothian). Most pharmacists interviewed were positive about the training they received and indicated that it prepared them to deliver the intervention as planned. Participants' accounts of their experiences within participating pharmacies suggest that fidelity of delivery was largely achieved, with most describing positive and informative encounters, although just over a quarter of intervention participants (54/198) could not recall being given a 'rapid access card' for an appointment at the study SRH clinic.[11] Those who attended SRH services described less positive experiences, including services being too busy and a lack of awareness among staff. For more detail on participants' experiences within the pharmacy and SRH context, and other relevant fidelity data, see online supplemental data file 6.

### Mechanisms of impact
#### Overcoming barriers to accessing routine contraception
Pharmacists', SRH clinical staff and participants' accounts suggest that bridging as a practice within pharmacies may have positive impacts on women's contraceptive awareness and use in the short, and potentially in the longer term. Many participants discussed how being approached within the pharmacy and being offered a bridging method acted as a necessary prompt to change contraceptive practices, as typified by Participant 10 (Lothian): 'It made me kind of realise that it was time to go on one and that it was something I did need to do'. This reinforces

enthusiasm from pharmacists within training sessions and during interviews for the EC consultation as an opportune moment to intervene, and how offering bridging could potentially disrupt repeat EC use, which was viewed as a persistent issue within some community pharmacies.

Many participants emphasised the pharmacy setting in particular as being pivotal to overcoming barriers faced in accessing contraception, some of them personal, including lack of time and embarrassment and some structural, such as difficulties accessing healthcare appointments within traditional settings:

> *I thought it was really good actually, because yes, usually it's like you have to make an appointment with your GP and maybe, like, if you live in a busy area it can be a couple of weeks that you have to wait, you know, so it was just quite nice being able to go into the pharmacy and, you know, get a longer term solution, if that makes sense* (Participant 17, Lothian)

Similarly, pharmacists and SRH clinical staff highlighted the accessibility and convenience of pharmacies as pivotal in overcoming such barriers, particularly for young people and students: 'A lot of people that actually say, yes, they've been wanting to go on contraception for a long time but they didn't have the time or they can't make the time to go to a sexual health clinic' (Pharmacist 20, London).

While ease of access seemed to be a key mechanism of impact, analysis of screening log data and pharmacist interview data did highlight the ingrained nature of such barriers, with lack of time and potential embarrassment noted as key barriers to participation in the study. Pharmacists discussed a sense of rush common to EC consultations, fuelled by embarrassment, which impacted on participation:

> *'I expect, embarrassment, that they just wanted to come in and out, you know, we are talking about something that people feel embarrassed about, they just want to come in, swallow the tablet, get out, forget the whole thing ever happened'* (Pharmacist 8, Tayside)

As well as issues of time and embarrassment, narratives of resistance within EC consultations to take the specific contraceptive offered, or hormonal contraception more generally were also commonly mentioned as barriers to participation: 'I had a few people who just didn't really like the sound of hormones' (Pharmacist 17, Tayside). Such barriers may have implications relating to wider uptake of this service within pharmacies, shedding insight into reasons why some chose not to participate in the study.

### Increased awareness, confidence and self-efficacy

Participants described other benefits of the intervention relating to the information provided within the pharmacy, including greater awareness of contraception and contraceptive services: 'I found out more about it [contraception]. I've got more knowledge of that type of stuff now so that's one of the positive things, I guess'

(Participant 9, London). For some, this increased awareness resulted in improved confidence in accessing and using contraception:

> *It's meant that I'm on the pill, I've got that sorted, I know that I can go to the pharmacy to get advice, I hopefully won't be needing the emergency contraception again, but I know that I can get it there if, for whatever reason, I need it. Yeah, I think, it's probably given me a bit more confidence with it as well.* (Participant 36, Tayside)

Participants' accounts drew attention to some of the mechanisms of change: viewing contraception as accessible, and increased awareness, confidence and self-efficacy, leading to potentially healthier behaviours and attitudes towards risk. This suggests that the intervention likely prompted participants to think more about their sexual health and longer-term contraception, as well as raising awareness of available contraceptive services.

### Facilitators of, and barriers to, continued uptake of routine contraception

It is important to shed light on why the intervention worked for some, and not for others. As reported within the outcomes paper,[11] more than half (112/198) of intervention participants were on effective contraception at 4-month follow-up, and 16 of the participants interviewed described being on POP, or another effective method, after recruitment into the study (including previous non-users and past-users with negative experiences on other forms of hormonal contraception). Those who remained on effective contraception tended to find the process of accessing further contraception from their GP/SRH clinic straightforward, and reported no obvious side effects from POP:

> *I don't feel that there has been any side-effects, like of like up and down moods or mood swings that some other women get on different pills, which is very positive* (Participant 1, Lothian).

Another facilitator of continued POP use seemed to be familiarity with oral contraception: 'At the moment I do feel happy on it and it's convenient, I'm used to taking the pill, and my friends are like, 'oh coil is so easy because you don't have to think about it', but I'm used to it' (Participant 18, Lothian).

While many participants had positive experiences of taking part in the Bridge-it study, and were on regular contraception at 4-month follow-up, just under half of all intervention participants were not on contraception at 4-month follow-up (n=88/198).[11] Data from the 4-month follow-up survey and participant interviews highlighted common reasons, including not being currently sexually active, side effects concerns, and difficulty arranging or finding the time to attend an appointment to access further contraception.[11] In particular, a quarter of intervention participants (n=40/158) discontinued POP due to side effects, with interview participants describing a range of adverse side effects experienced including

spotting, prolonged bleeding, skin problems, poor mental health and mood changes, headaches, weight gain, lowered libido and nausea. The most common side effect mentioned by interview participants were spotting and prolonged bleeding, typified by participant 12 (Lothian): 'There was blood every day and not much but enough to be annoying, if you know what I mean. So that's why I only took one packet and then I stopped because I was just like I can't'. Prior to taking part in the Bridge-it study, 22 of the interview participants attributed not being on contraception at entry to the study to previous negative contraceptive side effects, highlighting the persistent difficulties faced relating to well-being.

For some, not being able to continue accessing POP through the pharmacy acted as a barrier: 'And if I could just…because I don't want to have to book an appointment at the GP, you know…if I could just go to the pharmacy and get something I probably would have done it' (Participant 22, Lothian). As well as difficulties accessing appointments at GP/SRH clinics, participants' highlighted potential embarrassment and stigma related to attending SRH clinics as a barrier to the rapid access component of the intervention: 'I think I would rather go to the GP, but only because I feel like it is a little bit of a taboo to say I'm going to the sexual clinic' (Participant 27, Lothian). Consistent with these concerns, very few intervention participants attended their local SRH clinic (17%, n=52), and the majority who accessed more POP/alternatives did so via their GPs, suggesting that the incorporation of SRH clinics as an option for seeking ongoing contraception added little to the intervention.[11] While overcoming initial access barriers, participants' accounts highlight that providing a limited supply of POP from the pharmacy and offering rapid access to SRH services did not always succeed in overcoming long-term, recurring barriers to effective contraceptive use.

### Context
#### Participating pharmacies: competing priorities and staffing issues
A range of cross-cutting challenges to implementation of the intervention emerged. Pharmacists highlighted existing contextual challenges, such as high workloads, expanding roles, competing priorities and staff shortages: 'it never feels like you have enough people' (Pharmacist 12, Lothian). These existing challenges influenced delivery of the Bridge-it Study in practice, contributing to deprioritisation of participant screening at busy times and slow recruitment rates: 'there were a few times I possibly could have done an intervention but I didn't because I knew my queue was too big' (Pharmacist 7, Lothian). Pharmacists highlighted the added burden of the research context (eg, study paperwork) as well as the additional required Patient Group Direction (PGD) for the POP, extending EC consultations by approximately 15–20 min. However, pharmacists tended to be positive about embedding a bridging service within everyday practice: 'the paperwork aspect [research-related] doesn't fit in because it's quite time consuming, but the actual clinical aspect and the

reason behind it makes a lot of sense' (Pharmacist 14, London). While existing challenges and pressures related to services currently provided within pharmacies should be considered in wider implementation, the provision of bridging appeared to be feasible and acceptable within the community pharmacy context, with the majority of concerns typically related to the additional research burden of the intervention.

#### Participating SRH clinics: funding cuts and changing service provision
SRH clinical staff described continually trying to manage priorities to cope with staff shortages, funding cuts and changing service provision: 'You know, we're constantly trying to juggle, and constantly trying to desperately figure out if we take somebody off this clinic then maybe we could cover that clinic…' (SRH staff 1, Lothian). Accounts highlighted the reshaping of services to accommodate limited funding and resources, with two study sites moving to triaging of all patients, and from walk-in to priority access clinics. Most described an increased focus on young people's services, and a move away from routine contraception provision to a focus on more specialised services: 'Because obviously we were providing the more specialist stuff, whereas people that would be looking just for routine contraception would be encouraged to attend their GPs, rather than come to the specialist service, just because the lack of capacity' (SRH staff 3, Tayside). This had potential implications relating to the implementation of the Bridge-it Study, and concerns were raised relating to services having the resources to cope with rapid access, and the lack of fit with current practice priorities. Some worried that this may have resulted in Bridge-it participants being missed or turned away: 'And although the nurses were trying to get the information from patients if they had been involved in the Bridge-it study, if the patient did not specifically explain that they probably wouldn't have been able to get into the clinic that easily' (SRH staff 2 Tayside). Such concerns were founded, with some participants advised to instead attend their GP. A lack of fit with existing service provision may impact on implementation and raises issues around wider implementation in this format. Changing service provision, combined with lack of engagement with the rapid access component of the intervention suggests that signposting to SRH services may be sufficient and more realistic.

### DISCUSSION
#### Why did the intervention work?
The findings from this multisource process evaluation confirmed our hypothesis that providing access to effective bridging contraception through provision within community pharmacies and signposting to local contraceptive services facilitates uptake of ongoing effective contraception, as highlighted within the outcomes paper.[11] Positive impacts on participants' contraceptive practices were evident, with the convenience and

accessibility of pharmacies appearing to be pivotal in overcoming well-established access barriers to contraception.[18 19] This adds to the growing literature emphasising the accessibility of community pharmacies, and enthusiasm for the pharmacy as an option for contraceptive service provision.[8 16 20] The process evaluation shed light on other mechanisms of change highlighted in previous studies.[19 21 22] These included increased awareness of contraception and contraceptive services, motivation, and perceptions of self-efficacy, leading to potentially healthier behaviours and confidence in managing sexual risk-taking.

Despite existing challenges within the pharmacy and SRH provider context, bridging of POP as a practice within the community pharmacy setting seemed to be welcomed by pharmacists, SRH clinical staff and participants. Accounts emphasised the acceptability of the intervention and existing demand for pharmacy provision of routine contraception, indicating alignment of intervention design and patient need. This suggests that bridging of POP as a practice within community pharmacies is acceptable and feasible and has potential to be widely implemented and successfully embedded within routine practice. A lack of engagement with the rapid access component of the intervention and changing SRH service provision suggest that signposting to SRH services may be sufficient in wider implementation.

### How do we optimise wider implementation and improve outcomes?

As a result of the Bridge-it study trial success, bridging as a practice has been implemented within community pharmacies in Scotland.[12] It is vital to address implementation challenges, and work to alleviate persistent barriers to accessing and using effective contraception to optimise effectiveness and sustainability of the intervention in practice. To optimise uptake of bridging within the pharmacy context, it is important to acknowledge barriers to participation encountered, including lack of time, embarrassment, and lack of choice of bridging contraception offered, as well as existing contextual challenges within the pharmacy setting. The retail setting, lack of resources and expanding services emphasise the need for sufficient time and resources to administer bridging adequately to be embedded within routine 'everyday' practice. Recommendations to increase uptake of bridging contraception within the pharmacy setting include greater advertising of the service to raise awareness; flexibility regarding accessing routine contraceptive services within pharmacies (eg, option to book appointments) to overcome time-related barriers; maintenance of non-judgemental and supportive contraceptive consultations to alleviate embarrassment; and the need for future research into the feasibility of offering alternative contraceptive options within the pharmacy context for those resistant to taking POP specifically.

While the incorporation of bridging within the pharmacy setting in Scotland is a step forward in increasing access to longer-term contraception,[12] it is important to recognise that it is not a comprehensive solution, and acknowledge the potential limitations of this approach. The intervention did not work for all and persistent barriers to accessing and using effective contraception remain, echoed in previous literature,[18 19 23] including worries about side effects, ingrained stigma relating to accessing contraception particularly within SRH services, and difficulties accessing appointments for continued contraceptive care. Under current regulations, after provision of a bridging supply within community pharmacies, patients in Scotland are directed to their local GP practice or local SRH service for ongoing contraception.[12] Participants' experiences highlight that while bridging within the pharmacy context was key in overcoming initial access barriers to regular contraception, the need to access traditional contraceptive settings (eg, GP, SRH clinics) for ongoing contraception maintained barriers to continuation. For others, barriers to regular uptake of contraception were primarily well-being related, highlighting persistent difficulties faced in contraceptive journeys, and the need for a central focus on well-being within contraceptive consultations. Such challenges should be acknowledged in the design of future contraceptive service trials, and our key recommendations to increase uptake of ongoing contraception include: clear and consistent sign-posting of contraceptive services; key focus on well-being within contraceptive consultations; greater linkage with GP practices; easier processes for obtaining repeat supplies from the pharmacy without the need for a prescription, and consideration of longer-term contraceptive care within the community pharmacy context. Some of these recommendations could be relatively straightforward to implement (eg, continuing professional development course on supportive well-being led consultations), while others would require practice, regulation or policy change. The Scottish government has highlighted a commitment to provision of more routine sexual healthcare, including access to broader contraception services within the pharmacy context.[24] It is important to note that the findings from this study are specific to the UK context and implementation in other settings would require consideration of context-specific regulations and contraceptive availability.

### Strengths and limitations

Previous evaluations of interventions within the pharmacy context have often focused on exclusively quantitative measures.[25] In contrast, The Bridge-it process evaluation combined qualitative and quantitative methods to provide comprehensive and robust insights into implementation of the intervention, mechanisms of change and important contextual factors. There are limitations. As participants were followed up 4 months postintervention, and qualitative interviews were conducted at one time point, we are unable to confidently comment on continuation of the chosen contraceptive method and longer term implementation of the service. Due to practical reasons, direct

observation of pharmacist training sessions only took place at Scottish sites, and we were unable to observe implementation of the intervention in practice within the pharmacy and SRH context, making assessing fidelity difficult. In addition, lack of engagement with the rapid access component and difficulties recruiting SRH clinical staff for interview meant that accounts within the SRH context were limited. While purposive sampling was used to ensure the pharmacists and participants recruited for interview were diverse, it is possible that participants may have been more likely to agree to interview due to particularly positive or negative experiences of the study, and the generalisability of findings are limited to accounts from those who agreed to take part in the trial. It is important to acknowledge that participating pharmacies and pharmacists may be more positive about the intervention than those who did not wish to participate in the study (due to barriers such as existing workload). It should also be noted that pharmacists and participants were being asked to reflect on experiences up to 6 months previously, which may have impacted recall.

## Conclusion

Providing a bridging supply of the POP with EC from community pharmacies had positive impacts on contraceptive practices in the short term, and potentially in the longer term through overcoming some of the existing barriers to access and through increasing users' confidence in accessing contraception. The accessibility and convenience of the pharmacy setting was pivotal in making effective contraception more accessible. Implementation appeared to be acceptable, welcomed and feasible to be routinely embedded within pharmacy practice. Lack of engagement with the rapid access component of the intervention and changing SRH service provision suggest that sign-posting to SRH services may be sufficient. If widely implemented, provision of bridging contraception within community pharmacies has the potential to increase access to contraception and prevent more unintended pregnancies for women. Persistent challenges to ongoing contraceptive use should be considered in the design of future contraceptive service trials, and highlight the need for a package of solutions to ensure all needs are met.

**Author affiliations**

[1]MRC/CSO Social and Public Health Sciences Unit, University of Glasgow, Glasgow, UK

[2]Institute for Social Science Research, The University of Queensland, Brisbane, Queensland, Australia

[3]School of Social and Political Sciences, University of Glasgow, Glasgow, UK

[4]Edinburgh Clinical Trials Unit, Usher Institute, University of Edinburgh, Edinburgh, UK

[5]Department of Obstetrics and Gynaecology, University of Edinburgh, Edinburgh, UK

[6]Directorate of Public Health, NHS Tayside, Dundee, UK

[7]Division of Cardiovascular Medicines and Diabetes, Ninewells Hospital and Medical School, Dundee, UK

[8]Elizabeth Garrett Anderson Institute for Women's Health, University College London, London, UK

[9]King's College Hospital and King's Centre for Global Health and Health Partnerships, King's College London, London, UK

[10]Institute for Global Health, University College London, London, UK

[11]Sexual and Reproductive Health, NHS Lothian, Edinburgh, UK

**Acknowledgements** The authors would like to thank participating sites, the pharmacists, SRH clinical staff, and participants who took part in interviews, the Bridge-it Study Steering Committee, the Bridge-it Study Patient and Public Involvement Group, research staff involved in implementing the study and Katherine Lewis for excellent project support.

**Contributors** SP led on data collection and analysis, and on the initial draft of the manuscript. LM was a coinvestigator of the Bridge-it study, and led on the process evaluation, contributing to study design, supervising the project and its staff, and made significant contributions to drafting and revising the manuscript. KS contributed to data collection and analysis, and commented on and approved the final version of the manuscript. AG, AR and JMS were coinvestigators, and contributed to conception and design of the study, and commented on and approved the final version of the manuscript. CB, AJ, AM, DS and NS recruited participants, facilitated data collection, were responsible for day-to-day management of the main study within the three sites, and commented on and approved the final version of the manuscript. STC was the chief investigator and guarantor of the Bridge-it study, and contributed to conception and design of the study, and drafting and revising the manuscript.

**Funding** Funding for the research was provided by the National Institute for Health Research's Health Technology Assessment Programme. HTA Project: 15/113/01. LM and SP are funded by the UK Medical Research Council and Chief Scientist Office of the Scottish Government Health and Social Care Directorates at the MRC/CSO Social & Public Health Sciences Unit, University of Glasgow (MC_UU_12017/11, SPHSU11; MC_UU_00022/3, SPHSU18).

**Disclaimer** The views expressed are those of the author(s) and not necessarily those of the NIHR, the NHS or the Department of Health and Social Care.

**Competing interests** STC reports grants from the National Institute for Health Research (Health Technology Assessment (NIHR HTA) Programme), during the conduct of the study. AG is a consultant to HRA Pharma. AR reports receiving research grants from Gilead Sciences, Bristol-Myers Squibb, AbbVie and Roche; honorariums from Gilead Sciences; and personal fees from AbbVie. AM reports grants from NIHR HTA during the conduct of this study. AM is a clinical support bank midwife for SH:24 and a research midwife at Oxford University. All other authors declare no competing interests.

**Patient consent for publication** Not applicable.

**Ethics approval** This study involves human participants and was approved by This study was granted ethical approval by the South East Scotland REC in June 2017 (17/SS/0080). Approvals have been obtained from NHS Research Scotland (NRS) and Health Research Authority England (HRA) prior to commencement of the study.Protocol number AC17032IRAS project ID 208918 Participants gave informed consent to participate in the study before taking part.

**Provenance and peer review** Not commissioned; externally peer reviewed.

**Data availability statement** Data are available on reasonable request. Access to anonymised data are available on reasonable request. All data requests should be submitted to the Chief Investigator (STC) (Sharon.Cameron@ed.ac.uk) for consideration.

**ORCID iDs**
Susan Patterson http://orcid.org/0000-0003-1846-5749
Lisa McDaid http://orcid.org/0000-0002-7711-8723
Andrew Radley http://orcid.org/0000-0003-4772-2388
Alessandra Morelli http://orcid.org/0000-0002-9803-2136
Sharon Tracey Cameron http://orcid.org/0000-0002-1168-2276

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
