## [Reviewer comments · BMJ Open]

ARTICLE DETAILS

TITLE (PROVISIONAL)	Improving effective contraception uptake through provision of bridging contraception within community pharmacies: findings from The Bridge-it Study process evaluation
AUTHORS	Patterson, Susan; McDaid, Lisa; Saunders, Kristina; Battison, Claire; Glasier, Anna; Radley, Andrew; Stephenson, Judith; Johnstone, Anne; Morelli, Alessandra; Sally, Deirdre; Stewart, Nicola; Cameron, Sharon

VERSION 1 – REVIEW

REVIEWER	Marston, Cicely London School of Hygiene and Tropical Medicine, Department of Social and Environmental Health Research
REVIEW RETURNED	30-Sep-2021

GENERAL COMMENTS	This paper is a process analysis provided to supplement the Bridge-It trial findings. 1. The paper contains interesting information and the reader would benefit from a more explicit exposition of what the findings add to the published trial results and what the implications of the findings are for practice.2. The important change to POP availability from pharmacies since the trial is not yet mentioned. This should be added. In addition, it would be interesting to know how the data from this study might help us understand what to expect/how to improve outcomes from the new availability of POP from pharmacies.3. The aims of the study (p.6) balance questions of implementation and impact on contraceptive practice. To balance the findings accordingly, the paper would benefit from more information on contraceptive practices/context/uptake. If space is a problem, the detail on intervention fidelity could be reduced.a. relevance of the fidelity issues should be made more explicit - should we consider the trial results unreliable? Do the issues tell us something about how to improve ongoing practice? If the answer to both is 'no' then moving some of the detail to an appendix would provide information to inform future studies leaving more space in the main text. Some specific comments: 4. Abstract: "Outcome measures". It is unclear how a mechanism of change or a 'contextual factor' can be considered an 'outcome measure'. Later reference to 'outcome data' (p.8 line 5) seems to refer to the trial outcome data rather than the outcome measures set out here. Suggest rephrasing for clarity and consistency.
---

5. Introduction - first paragraph - 'currently the majority of community pharmacies are only able to provide condoms without a prescription' - while this was the case at the time of the study, changes since then should be mentioned to avoid confusion. The authors say 'Findings will contribute to informing translation into practice, shedding light on how wider implementation of the service within community pharmacies may be optimised' (p.4, line 16). This seems likely and so in the discussion it would be useful to spell out some specific contributions/recommendations/implications.

6. Some more information about the interviews would be helpful to understand the data generation process. E.g. How long were they? How were they carried out (e.g. were they a series of fairly structured questions or more free form?) How were interviewees selected? Were client interviewees broadly typical of pharmacy EC clients or were they more/less diverse? Assuming the numbers relate to sites, almost all client interviewees were in Scotland (9 in London, 24 in Edinburgh, 3 in Dundee). This, or any other characteristic of the participants, is not discussed - would we expect a different response in Scotland compared with England? What ages were the participants in the interviews? Was there any diversity with respect to ethnicity? There is a brief reference to 'diverse' participants and it would be useful to know what form this diversity took or its implications for the findings/how to understand intervention mechanisms/response.

7. There is reference to 'independent analysis of each data strand' - what is a data strand? Is this a reference to the different sources of data or perhaps it is a reference to quantitative vs qualitative data?

8. P.6 line 42 - How were the themes arrived at? There is reference to thematic analysis and framework analysis - how did these differ?

9. Author positionality: the paper does not yet contain any comment on reflexivity/positionality (currently indicated as 'not reported' in the SRQR checklist. This needs to be added, and the checklist completed accordingly). For instance:

- Who conducted the interviews?
- Did the funder play any role in the intervention or evaluation design?
- There is reference to 'a research assistant' who was this? Is it one of the authors?
- The sections on implementation provide a clear picture of the pressures pharmacists are under that might limit their ability to intervene in this way. It would be helpful if the authors would comment on the extent to which the participating pharmacies might be expected to be more positive than those who did not wish to participate in the study, and the extent to which any prior relationships between interviewers and pharmacists via the trial might have affected the reporting at interview.

10. PPI - were specific members of the PPI group involved in the process evaluation? They do not appear in the acknowledgements - is this an oversight or perhaps they were not specifically involved in this part of the study? Who were the PPI group in terms of expertise relevant to the project? Were they EC users?

11. There is information on workload for pharmacists and how this might have affected the intervention - do the authors think this might have affected the final results from the trial?

12. Were clients asked at qualitative interview about their reasons for participating and for continuing or discontinuing POP? More information on this (or similar) would help rebalance the paper to contain more on uptake. For instance, was there any information

	from the qualitative data on whether the reporting of contraception use at 4 months in the survey referred to 'I am using a method today' vs 'I used a method last time I had sex' vs 'I had a break then went back to the POP so am now using it again' (or a mix of these?). The survey data show discontinuation attributed to side effects - do we know from the qualitative interviews whether these were side effects that were being experienced or side effects that were feared? The interview data seem to have the opposite (i.e. that there were no side effects) - it seems possible that people who were happy with the method would be more likely to agree to do an interview - or were there methods of sampling that would make this less likely? Small editing points: P.6 line 5 'desire among women' - this needs to be expanded a little more - which women? p.12 line 16 "many participants... overcame barriers to contraceptive uptake" - what barriers were they and how did the participants overcome them? Is this a reference to the 'lack of time, and avoidance' (p.10 line 51)? Here 'avoidance' also needs further explanation, perhaps with an illustrative example from the data to clarify. P.12, line 7 - quote is not italicised. The findings and interpretation are mixed in the write up. For clarity, the findings should be reported in the findings section and then the interpretation in the discussion section (e.g. p.10 line 27 onwards is interpretation).
--	--

REVIEWER	Fagbamigbe, Adeniyi University of Ibadan, Epidemiology & Medical Statistics
REVIEW RETURNED	21-Oct-2021

GENERAL COMMENTS	This is an interesting study to evaluate the implementation process. It addresses a significant public health challenge. It is uncommon to have such studies in scientific settings. It is definitely a good way to know if the intervention is on course and the possibility of achieving its goal. A key strength of the study is the combination of qualitative and quantitative methodologies to provide comprehensive and robust insights into implementation of the intervention, mechanisms of change, and important contextual factors. I only have few comments. Abstract L37: from which result in the abstract did you draw this conclusion? P4 L8: in “,,with abortion rates in 2018 reaching the highest numbers recorded since 2008...”can you please provide the rate per women Page 6 L 12 One of the study aim was to understand “How did the delivered intervention impact on contraceptive practices?” In the design, was contraceptive practices an output or an outcome? One of them does not qualify to be part of the process. A good evaluation framework would guide your readers The evaluation framework (EF) is missing. You can not evaluate without a provision of an EF. You should construct any of theory of change, logic model, result framework or logframe.
--

	Page 7 L6: “All data were analysed independently of the outcome data,” ...I don’t get this Page 14: L18: the authors asked “Why did the intervention work?” did you evaluate the intervention or its implementation? This are two different things....Can you evaluate the intervention just few months/years after implementation? To answer that question you need a passage of time. That should be the evaluation of impact of the intervention. Again, there is no evaluation framework to contextualize the specific goals. So far your goals are to understand:  • Was the intervention implemented as planned? • How did the delivered intervention impact on contraceptive practices? • How did the local and broader context affect implementation and outcomes? These are not about whether intervention worked or not. An intervention may be successful at implementation but not at the delivery of the goal. That’s what differentiate intervention effectiveness and efficiency. Note: Efficiency is defined as the ability to accomplish something with the least amount of wasted time, money, and effort or competency in performance. Effectiveness is defined as the degree to which something is successful in producing a desired result; You have evaluated the efficiency, not effectiveness, so you cant determine if the intervention worked or not. Else, the words “process evaluation” is misused.
--	---

REVIEWER	Black , Kirsten University of Sydney
REVIEW RETURNED	22-Oct-2021

GENERAL COMMENTS	Thank you for your process evaluation of the Bridge-it study. It highlights some important issues and assists with understanding the ways in which the trial succeeded and the ways in which barriers to contraceptive access were not dismantled. I recommend the authors consider a few minor changes for clarity/improvement Abstract This sentence is not clear to me. Page 3 line 47-49 “However, lack of engagement with the rapid access component of the intervention and existing SRH context suggest that signposting to SRH services may be suffice Introduction In the section describing the main study it would be helpful to add the time frame (ie 4 months within these sentences) Page 4 line 49-53 “Analysis of the main outcome of the study demonstrated a greater proportion of women using effective contraception within the intervention group (58.4% SD 21.6) compared to the control group (40.5% SD 23.8)[9]. Full details on the trial protocol and outcomes are reported elsewhere[9,10]” Results The results report that at the 4 month interview 1/4 of women did not recall receiving the rapid access card but 90% recorded receiving information on contraception. The authors comment on
---

	fidelity and I wonder if they thought it worth tempering their comments that there was high fidelity to the intervention given the first result (backed up by a qualitative quote). Discussion I think it would be worthwhile mentioning the applicability to other settings as well as the fact that you have available in the UK a POP that inhibits ovulation so more suitable for ongoing contraception in highly fertile young people.
--	--

VERSION 1 – AUTHOR RESPONSE

Reviewer 1 comments:

Comment	Response
1. The paper contains interesting information and the reader would benefit from a more explicit exposition of what the findings add to the published trial results and what the implications of the findings are for practice.	To highlight more explicitly what the findings add to the trial results and our recommendations for practice, we have as suggested, reduced the detail on intervention fidelity, provided more information on contraceptive practices, uptake and context within the results section (Pg 9, lines 13-27, Pg 10 lines 1-19; pg 11 lines 26-32) and have developed our discussion of recommendations for wider implementation (Pg 14-15).
2. The important change to POP availability from pharmacies since the trial is not yet mentioned. This should be added. In addition, it would be interesting to know how the data from this study might help us understand what to expect/how to improve outcomes from the new availability of POP from pharmacies.	We have added text detailing the recent changes to contraceptive provision within community pharmacies (Pg 4, lines 14-19), and have developed our discussion of implications for practice (Pg 14-15).
3. The aims of the study (p.6) balance questions of implementation and impact on contraceptive practice. To balance the findings accordingly, the paper would benefit from more information on contraceptive practices/context/uptake. If space is a problem, the detail on intervention fidelity could be reduced. a. relevance of the fidelity issues should be made more explicit - should we consider the trial results unreliable? Do the issues tell us something about how to improve ongoing practice? If the answer to both is 'no' then moving some of the detail to an appendix would provide information to inform future studies leaving more space in the main text.	In order to balance the findings more accordingly, we have reduced the text on fidelity issues (included as supplementary data file 6) and have provided more information on contraceptive practices, uptake and context within the results section (Pg 9, lines 13-27, Pg 10 lines 1-19; pg 11 lines 26-32). This has helped us to include more information on recommendations for wider implementation and future studies (Pg 14-15).
4. Abstract: "Outcome measures". It is unclear how a mechanism of change or a 'contextual factor' can be considered an 'outcome measure'. Later reference to 'outcome data' (p.8 line 5) seems to refer to the trial outcome data rather than the outcome measures set out here. Suggest rephrasing for clarity and consistency.	To address this comment, we have reorganised the abstract into the following headings: objective, research design and methods, results and conclusions. These abstract headings are consistent with previous process evaluations published in BMJ Open.
5. Introduction - first paragraph - 'currently the majority of community pharmacies are only able	We have added text detailing the recent changes to contraceptive provision within

to provide condoms without a prescription' - while this was the case at the time of the study, changes since then should be mentioned to avoid confusion. The authors say 'Findings will contribute to informing translation into practice, shedding light on how wider implementation of the service within community pharmacies may be optimised' (p.4, line 16). This seems likely and so in the discussion it would be useful to spell out some specific contributions/recommendations/implications.	community pharmacies (Pg 4, lines 14-19), and have developed our discussion of recommendations for wider implementation (Pg 14-15).
6. Some more information about the interviews would be helpful to understand the data generation process. E.g. How long were they? How were they carried out (e.g. were they a series of fairly structured questions or more free form?) How were interviewees selected? Were client interviewees broadly typical of pharmacy EC clients or were they more/less diverse? Assuming the numbers relate to sites, almost all client interviewees were in Scotland (9 in London, 24 in Edinburgh, 3 in Dundee). This, or any other characteristic of the participants, is not discussed - would we expect a different response in Scotland compared with England? What ages were the participants in the interviews? Was there any diversity with respect to ethnicity? There is a brief reference to 'diverse' participants and it would be useful to know what form this diversity took or its implications for the findings/how to understand intervention mechanisms/response.	As requested, further details have been added relating to the interviews, including: average interview time for each group (Pg 6, line 32-33, pg 7 lines 8-9, lines 13-18); information on topic guides and type of interview (Pg 5, Lines 30, Supplementary data file 1); process of interviewee selection and sampling strategy (Pg 6, lines 20-34; Pg 7 lines 5-9, Pg 7 lines 14-17) ; greater detail on participant characteristics (Pg 6, lines 20-33); and explanation of greater number of participants in Lothian (Pg 6, Lines 29-34).
7. There is reference to 'independent analysis of each data strand' - what is a data strand? Is this a reference to the different sources of data or perhaps it is a reference to quantitative vs qualitative data?	'Data strand' refers to each source of data – we have reworded to make this clearer (Pg 8, lines 9-17).
8. P.6 line 42 - How were the themes arrived at? There is reference to thematic analysis and framework analysis - how did these differ?	Further information has been added on how themes were generated using framework analysis (Pg 6, lines 8-16). We refer to thematic analysis when discussing the observational data where we did not incorporate a matrix into the analysis, but instead wrote up descriptive summaries guided by the observation proforma. Information has been added to make this clearer (Pg 7, lines 31-32).
9. Author positionality: the paper does not yet contain any comment on reflexivity/positionality (currently indicated as 'not reported' in the SRQR checklist. This needs to be added, and the checklist completed accordingly). For instance: a. Who conducted the interviews? b. Did the funder play any role in the intervention or evaluation design? c. There is reference to 'a research assistant' who was this? Is it one of the authors? d. The sections on implementation provide a clear picture of the pressures pharmacists are	We have added more information on author positionality, including who conducted the interviews (pg 5 lines 30-31), the role of the funder in intervention or evaluation design (pg 5, line 32), prior relationships between interviewers and pharmacists (Pg 5 lines 30-31, pg 6, line 1-2), and have included within the strengths and limitations section the importance of acknowledging that participating pharmacies and pharmacists may be more positive about the intervention than those who did not wish to participate (pg 16, Line 1-7).

under that might limit their ability to intervene in this way. It would be helpful if the authors would comment on the extent to which the participating pharmacies might be expected to be more positive than those who did not wish to participate in the study, and the extent to which any prior relationships between interviewers and pharmacists via the trial might have affected the reporting at interview.	
10. PPI - were specific members of the PPI group involved in the process evaluation? They do not appear in the acknowledgements - is this an oversight or perhaps they were not specifically involved in this part of the study? Who were the PPI group in terms of expertise relevant to the project? Were they EC users?	Service users provided insight into the design of the process evaluation through reviewing and commenting on study documentation. This has been added to the section on patient and public involvement (p 8), and to the acknowledgements.
11. There is information on workload for pharmacists and how this might have affected the intervention - do the authors think this might have affected the final results from the trial?	While there is potential that pharmacist workload may have impacted on participation, we are confident that the robust trial design mitigates any impact of this on the outcomes of the study. It is also important to note that the study design was shaped by learning from a previous pilot study, and this larger trial confirmed the pilot study findings within a more generalised setting.
12. Were clients asked at qualitative interview about their reasons for participating and for continuing or discontinuing POP? More information on this (or similar) would help rebalance the paper to contain more on uptake. For instance, was there any information from the qualitative data on whether the reporting of contraception use at 4 months in the survey referred to 'I am using a method today' vs 'I used a method last time I had sex' vs 'I had a break then went back to the POP so am now using it again' (or a mix of these?). The survey data show discontinuation attributed to side effects - do we know from the qualitative interviews whether these were side effects that were being experienced or side effects that were feared? The interview data seem to have the opposite (i.e. that there were no side effects) - it seems possible that people who were happy with the method would be more likely to agree to do an interview - or were there methods of sampling that would make this less likely?	We have added more information on uptake, including more detail on reasons for continuing or discontinuing POP. We have rewritten some of this section to make clearer that experienced side-effects were a major reason for discontinuation amongst interview participants (Pg 9, lines 13-27, Pg 10 lines 1-19; pg 11 lines 26-32)
P.6 line 5 'desire among women' - this needs to be expanded a little more - which women?	We have added more detail to make clear we are referring to women presenting for EC at community pharmacies (Pg 5, Line 13).
p.12 line 16 "many participants... overcame barriers to contraceptive uptake" - what barriers were they and how did the participants overcome them? Is this a reference to the 'lack of time, and avoidance' (p.10 line 51)? Here 'avoidance' also needs further explanation,	More detail has been added in this section to illustrate barriers to contraceptive uptake and ways in which these were overcome (Pg 9, lines 13-27, Pg 10 lines 1-19; pg 11 lines 26-32). We have removed avoidance, and changed to embarrassment which seems more suitable.

perhaps with an illustrative example from the data to clarify.	
P.12, line 7 - quote is not italicised.	Edited.
The findings and interpretation are mixed in the write up. For clarity, the findings should be reported in the findings section and then the interpretation in the discussion section (e.g. p.10 line 27 onwards is interpretation).	We have removed any interpretation from the results section, and have incorporated within the discussion section where applicable.

Reviewer 2 comments:

Comment	Response
Abstract L37: from which result in the abstract did you draw this conclusion?	The results section of the abstract has been edited to include information relating to this conclusion (pg 2, line 17-18).
P4 L8: in “,,with abortion rates in 2018 reaching the highest numbers recorded since 2008....” can you please provide the rate per women	We have added the rates per women: 12.9 per 1,000 women (aged 15-44) in Scotland [1]; 17.4 per 1000 women (aged 15-44) in England and Wales (Pg 4, lines 3-4).
Page 6 L 12 One of the study aim was to understand “How did the delivered intervention impact on contraceptive practices?” In the design, was contraceptive practices an output or an outcome? One of them does not qualify to be part of the process. A good evaluation framework would guide your readers	We have edited the evaluation framework to help guide readers (see Figure 1), which highlights our intermediate outcomes (e.g. improved knowledge and competence, increased intentions to use POP and services) and end outcomes (e.g. uptake of effective contraception).
The evaluation framework (EF) is missing. You can not evaluate without a provision of an EF. You should construct any of theory of change, logic model, result framework or logframe.	The evaluation framework (EF) has been added as a figure within the main text (Figure 1).
Page 7 L6: “All data were analysed independently of the outcome data,” ...I don't get this	Typically within process evaluations, the process evaluation data is analysed before the outcome of the trial is known to minimise bias in interpretation - we have edited the text to clarify this (pg 8, Line 9-12).
Page 14: L18: the authors asked “Why did the intervention work?” did you evaluate the intervention or its implementation? This are two different things....Can you evaluate the intervention just few months/years after implementation? To answer that question you need a passage of time. That should be the evaluation of impact of the intervention. Again, there is no evaluation framework to contextualize the specific goals. So far your goals are to understand:  • Was the intervention implemented as planned? • How did the delivered intervention impact on contraceptive practices? • How did the local and broader context affect implementation and outcomes? These are not about whether intervention worked or not. An intervention may be	The effectiveness of the Bridge-it trial was reported within the outcomes paper published in the Lancet (referenced in this paper where relevant: 9 Cameron et al 2020). This current paper complements the outcomes paper through reporting the findings of the process evaluation, providing insights into implementation processes and mechanisms of impact to help understand the overall study outcomes, and on how/why the intervention was effective (or not). We have edited the text within the introduction to make this clearer (Pg 4 line 17-18, pg 5 lines 1-4, lines 10-14), have edited the evaluation frame work (Figure 1), and have acknowledged the short-time frame within the limitations section within the discussion (Pg15 L26-29).

successful at implementation but not at the delivery of the goal. That's what differentiate intervention effectiveness and efficiency. Note: Efficiency is defined as the ability to accomplish something with the least amount of wasted time, money, and effort or competency in performance. Effectiveness is defined as the degree to which something is successful in producing a desired result; You have evaluated the efficiency, not effectiveness, so you cant determine if the intervention worked or not. Else, the words "process evaluation" is misused.	
--	--

Reviewer 3 comments:

Comment	Response
This sentence is not clear to me. Page 3 line 47-49 "However, lack of engagement with the rapid access component of the intervention and existing SRH context suggest that signposting to SRH services may be suffice	To clarify, we have edited the sentence to highlight we are referring to barriers within the SRH context (Abstract, Line 18, 23-24), and have added more detail within the results section of the abstract about the rapid-access component (Abstract, Line 17-18).
Introduction In the section describing the main study it would be helpful to add the time frame (ie 4 months within these sentences) Page 4 line 49-53 "Analysis of the main outcome of the study demonstrated a greater proportion of women using effective contraception within the intervention group (58.4% SD 21.6) compared to the control group (40.5% SD 23.8)[9]. Full details on the trial protocol and outcomes are reported elsewhere[9,10]"	We have added information on the time frame as requested (pg 5 line 2)
Results The results report that at the 4 month interview 1/4 of women did not recall receiving the rapid access card but 90% recorded receiving information on contraception. The authors comment on fidelity and I wonder if they thought it worth tempering their comments that there was high fidelity to the intervention given the first result (backed up by a qualitative quote).	We agree with the reviewer, and have tempered our fidelity conclusions, highlighting that the intervention was mostly delivered as intended but that there were some inconsistencies reported in delivery.
Discussion I think it would be worthwhile mentioning the applicability to other settings as well as the fact that you have available in the UK a POP that inhibits ovulation so more suitable for ongoing contraception in highly fertile young people.	We have added a statement highlighting that the findings are specific to the UK context and implementation in other settings would require consideration of context-specific regulations and contraceptive availability (p17, lines 3-5).

VERSION 2 – REVIEW

REVIEWER	Black , Kirsten University of Sydney
REVIEW RETURNED	03-Jan-2022
GENERAL COMMENTS	Thank you for addressing the issues raised. I think the changes have improved the clarity of this manuscript describing the process evaluation.